# Controlling nanowire growth through electric field-induced deformation of the catalyst droplet

Federico Panciera[1,2], Michael M. Norton[3], Sardar B. Alam[4], Stephan Hofmann[1], Kristian Mølhave[4] & Frances M. Ross[2]

Semiconductor nanowires with precisely controlled structure, and hence well-defined electronic and optical properties, can be grown by self-assembly using the vapour–liquid–solid process. The structure and chemical composition of the growing nanowire is typically determined by global parameters such as source gas pressure, gas composition and growth temperature. Here we describe a more local approach to the control of nanowire structure. We apply an electric field during growth to control nanowire diameter and growth direction. Growth experiments carried out while imaging within an *in situ* transmission electron microscope show that the electric field modifies growth by changing the shape, position and contact angle of the catalytic droplet. This droplet engineering can be used to modify nanowires into three dimensional structures, relevant to a range of applications, and also to measure the droplet surface tension, important for quantitative development of strategies to control nanowire growth.

[1] Department of Engineering, University of Cambridge, Cambridge CB3 0FA, UK. [2] IBM T.J. Watson Research Center, Yorktown Heights, New York 10598, USA. [3] Department of Mechanical Engineering and Applied Mechanics, University of Pennsylvania, Philadelphia, Pennsylvania 19104, USA. [4] Department of Micro and Nanotechnology, Technical University of Denmark, Kongens Lyngby DK-2800, Denmark. Correspondence and requests for materials should be addressed to S.H. (email: sh315@cam.ac.uk) or to K.M. (email: Kristian.Molhave@nanotech.dtu.dk) or to F.M.R. (email: fmross@us.ibm.com).

A versatile approach to the formation of nanostructures is growth by the vapour–liquid–solid (VLS) mechanism[1–4]. Unlike conventional thin film or bulk crystal growth, VLS nanowire growth relies on the presence of a liquid droplet to catalyse incorporation of the growth material, which is supplied from the gas phase. Growth takes place only at the catalyst/nanowire interface to form an elongated crystal structure or nanowire. The chemical composition, diameter, growth direction and even crystal structure of the growing nanowire are modulated by changes in the basic growth parameters of temperature, and source gas pressure and composition[5–11]. *In situ* experiments have shown that these parameters control the droplet geometry and composition[6,12,13]. Thus, the droplet has a fundamental role in determining the structure of the nanowire. The remarkable range of structures enabled by VLS can be thought of as the result of engineered changes to the droplet. For example, varying the droplet composition controls the composition of the deposited material, forming heterostructures[9] and embedded nanocrystals[14]. Changing the droplet contact angle by varying its volume can alter the nanowire diameter[5,15] and sidewall structure[12], and even, in some materials, switch between growth of one crystal structure and another[6]. Finally, changes in droplet position at the nanowire tip are connected to phenomena of nanowire kinking[16–18].

Droplet engineering has conventionally involved changing the growth temperature or the pressure(s) of the precursor gas(es). Here we describe a different approach to droplet engineering, application of an external electric field during growth. Applying an electric field deforms the droplet, directly altering its shape, contact angle and position, without affecting other aspects of growth. In contrast, pressure and temperature have multiple effects and nanowire growth often has a complex dependence on these parameters. We explore electric field effects by growing Si nanowires inside a microfabricated growth system that can be operated in an *in situ* TEM (transmission electron microscope). Two separate substrates form a capacitor in which nanowires grow while under observation in the TEM. We image the changes in droplet geometry and nanowire growth as voltage is applied. We show that response to the electric field is rapid, compared with the response to pressure or temperature changes. The field breaks the symmetry of nanowire growth, suggesting opportunities to create new types of complex, three dimensional structures. Furthermore, diameter modulation is possible and growth can be stabilized when nanowire kinking is not desired. Such experiments also probe the surface tension of the droplet. This parameter enters in all models describing VLS growth[15,19], but has only been measured on a macroscopic scale and in environments far from growth conditions[20,21]. We discuss how the results can be adapted for conventional reactors. VLS growth in an electric field in principle allows new opportunities for modulating growth and access to part of the parameter space that is otherwise unavailable, and can thus be a powerful tool for nanostructure control.

## Results

**Nanowire growth in an electric field.** In order to apply an electric field during VLS Si nanowire growth, we use the experimental design shown in Fig. 1a. A capacitor is formed by two loop shaped monocrystalline silicon cantilevers, each of which can be Joule heated separately by direct current[22]. The cantilevers are spaced a few micrometers apart and insulated from each other to allow up to $\pm 200$ V to be applied between them. Note that the highest voltage possible in these experiments is still well below the threshold for the breakdown of typical VLS Si precursor gases such as silane[23]. The sidewalls are {111} so that

nanowires that nucleate epitaxially can grow perpendicular to the electrodes. Nanowires are grown only on one loop and growth is stopped when one nanowire tip approaches within a few hundred nanometers of the opposite cantilever, referred to as the counter electrode (CE) (Fig. 1b). At this distance, applying 100 V between nanowire and CE (an electric field of $\sim 1$ V nm$^{-1}$) is sufficient to deform the catalyst droplet (Fig. 1c). Real-time imaging shows a rapid response of the droplet to the field (Supplementary Movie 1). To analyse the droplet deformation, we parametrize it by measuring the droplet aspect ratio, that is, the ratio between the semi-axes of the ellipse that best fits the droplet shape (Fig. 1b). Data collected over several voltage cycles show that the droplet deforms reproducibly (Fig. 1d) with an aspect ratio that depends on the magnitude but not the sign of the voltage (Fig. 1d inset). Note that at zero field the aspect ratio is $>1$, indicating that the droplet is a few per cent elongated compared with the expected spherical cap shape. We suggest that this deviation from spherical geometry is generated by the faceted nature of the nanowire on which the droplet sits. Si nanowires typically have a trigonal hexagonal cross-section, that is, alternating longer and shorter edges. A droplet pinned on such a base is expected to be non-spherical with different contact angles on opposite sides[24]. Although we cannot measure our nanowire cross sections directly, asymmetrical contact angles are evident in images, supporting this interpretation.

We find that field-induced deformation can affect nanowire growth in different ways. A field along the nanowire axis can modulate the diameter or stabilize growth, while a field at an angle can kink the nanowire. Figure 2a and Supplementary Fig. 1 show how a nanowire's diameter is changed, here by a factor of almost 2, by a strong axial field during growth that changes the droplet-nanowire contact angle. Since deformation is reversible, it is possible to modulate the diameter (See Supplementary Fig. 2). Figure 2b illustrates how an axial field stabilizes the growth direction during drastic changes of temperature and pressure. It is well known[7,16,25] that reducing temperature and pressure tend to destabilize growth because the droplet depins from the nanowire tip and wets the sidewall (Fig. 2b, second image), so that, as growth restarts, the nanowire kinks to one of the three equivalent ⟨111⟩ directions. However, a field applied before restarting growth repositions the droplet to its original position and prevents kinking. This mechanism may explain the improvement in alignment of nanowire forests grown under an electric field[26].

Figure 2c and Supplementary Movie 2 demonstrate the use of an off-axis electric field to induce kinking controllably. The nanowire shown grew initially in a ⟨111⟩ direction at an angle of $\sim 70^{\circ}$ to the field direction (defined by the sides of the capacitor). The off-axis field breaks the symmetry between contact angles on each sidewall. The contact angle opposite the CE is reduced and as a result the nanowire tapers and the droplet is squeezed onto an increasingly narrow pedestal. Eventually the droplet depins and wets the sidewall to relieve the force on both trijunctions[17]. The nanowire/droplet interface is now composed of two {111} facets (arrowed in the third image). Further growth shrinks the original facet and increases the new one until the growth direction changes completely (fourth image). Additional images and the final geometry of the nanowire and CE are shown in Supplementary Fig. 3. Several methods have already been proposed to induce nanowire kinking, such as temperature or pressure changes (including growth interrupts)[7,25] or changes in the gas environment[27]. However, these methods cannot select which of the symmetry-equivalent directions the nanowire will kink towards. Electric field-induced kinking appears to force the nanowire into whichever crystallographically preferred direction (⟨111⟩ for Si nanowires) is closest to the field direction. Figure 2d shows a variant of this concept. Instead of using a large, flat CE to

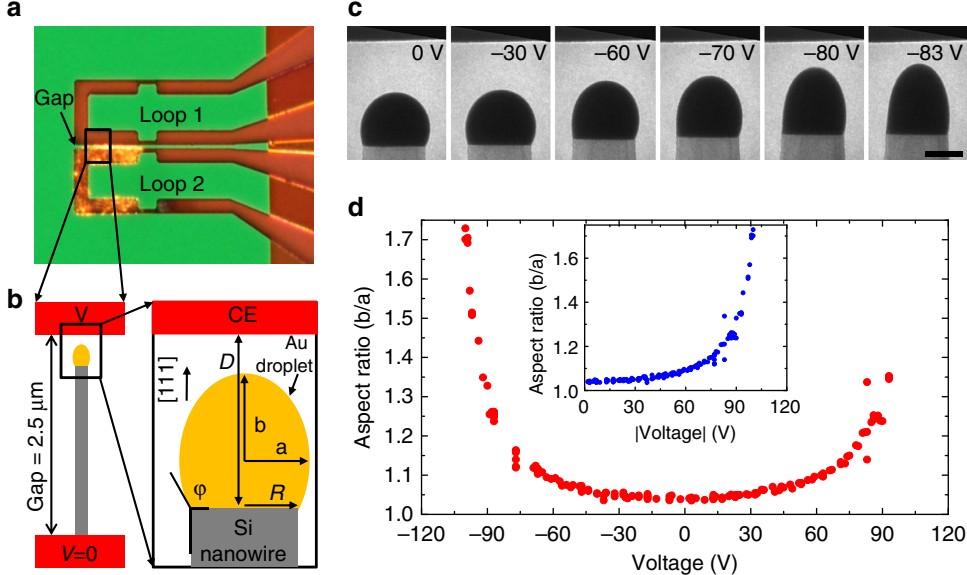

**Figure 1 | Experimental setup. (a)** Image recorded in a light microscope showing a microfabricated device composed of two Si loops that can be heated and biased independently. After depositing Au on the loops, a current of 1 mA passing through loop 2 heated the tip to 500 °C, allowing nanowire growth when disilane was flowed. The gap between loops is 2.5 μm. **(b)** Schematic showing a nanowire growing in the [111] direction across the gap, and parameters measured on each video image. Loop 2 is grounded and voltage is arbitrarily set at $V = 0$. **(c)** Sequence of video images showing a droplet (W1, see Fig. 4 below) deforming at ∼500 °C and the voltages shown. Scale bar, 100 nm. **(d)** Droplet aspect ratio ($b/a$) versus voltage $V$. Inset is $b/a$ versus the absolute value of $|V|$. Note the aspect ratio of 1.05 at zero field, discussed in the text.

set the field direction, one can use a nanostructure. A potential applied between this nanostructure and the growing nanowire can distort the droplet towards the nanostructure (third image), biasing nanowire growth towards the nanostructure. The ability to apply fields in different directions could potentially produce nanowires with specified three-dimensional structures.

**Measurement of surface tension.** The *in situ* experiments in Figs 1 and 2 have shown that nanowire growth is highly sensitive to distortions in the droplet shape. In order to use this droplet engineering to control nanowire morphologies in conventional growth reactors, it is important to be able to calculate the droplet geometry in different externally imposed field conditions. This requires knowledge of the balance between surface tension and local field, since surface tension tends to make the droplet spherical and the field typically elongates it. However, the surface tension of AuSi is not well known for the conditions (temperature, pressure and droplet dimensions) relevant to nanowire growth. We therefore analyse deformation as a function of field to obtain a measurement of droplet surface tension during growth.

In order to obtain a large data set and increase the measurement accuracy, we interrupted the growth of a nanowire at three different distances from the CE by decreasing the $Si_2H_6$ pressure (Fig. 3a). We then obtained deformation versus voltage (Fig. 3b,c). This involved parameterizing the droplet–vacuum interface as an ellipse (Fig. 4a). An ellipse appears to give a reasonable representation of the interface for points that do not lie near the droplet/nanowire interface, where the droplet is asymmetrically deformed. We use the ellipse parameters to construct an axisymmetric domain (Fig. 4b) on which the electric potential is calculated by solving the Laplace equation (see Methods). The droplet and the nanowire are considered to be perfectly conductive. After determining the electric field distribution along the droplet surface, we use it to

find the Maxwell stress distribution, which modifies the Young-Laplace equation:

$$\gamma\left(\frac{1}{R_1} + \frac{1}{R_2}\right) = \Delta P_0 + \frac{1}{2}\varepsilon_0 E_n^2, \qquad (1)$$

where $\gamma$ is the droplet surface tension, $R_1$ and $R_2$ the two position-dependent curvature radii, $\Delta P_0$ the internal pressure of the droplet, $\varepsilon_0$ the vacuum permittivity and $E_n$ the position dependent component of the electric field normal to the droplet surface. This equation is solved numerically by representing the axisymmetric surface as a curve parameterized by arc length, as described in Methods and in Bateni *et al.*[28] and Harris *et al.*[29]. Finally, comparing the calculated droplet shape with the TEM images yields the best fitting surface tension for each frame (Fig. 4c).

Figure 4d shows the aspect ratio versus $E$ curves from the three data sets in Fig. 3. The data fall on the same curve, implying that the three distances $D_x$ were accurately measured (see Supplementary Note 1). Figure 4e shows the surface tension values $\gamma$ extracted from these and another data set. Our solutions to the modified Young–Laplace equations assume axial symmetry of the base and droplet, hence predict that the droplet approaches a spherical cap geometry at zero field. Since, as discussed above, the droplet shape is not a spherical cap at zero field, our fits, therefore, yield values of $\gamma$ that tend to zero in weak fields to compensate (a weak field can only distort a droplet if it has low $\gamma$, see Methods).

The resulting surface tension value, $0.55 \pm 0.1 \, J \, m^{-2}$, is estimated by using the error weighted average of the data in Fig. 4e. This is significantly lower than literature values ($0.98 \, J \, m^{-2}$ (ref. 20) and $0.780 \, J \, m^{-2}$ (ref. 21)), perhaps due to the conditions (different temperature, lower pressure, nanometer scale droplets) and measurement method used here. It is also worth noting that $\gamma$ does not appear to depend on field, at least

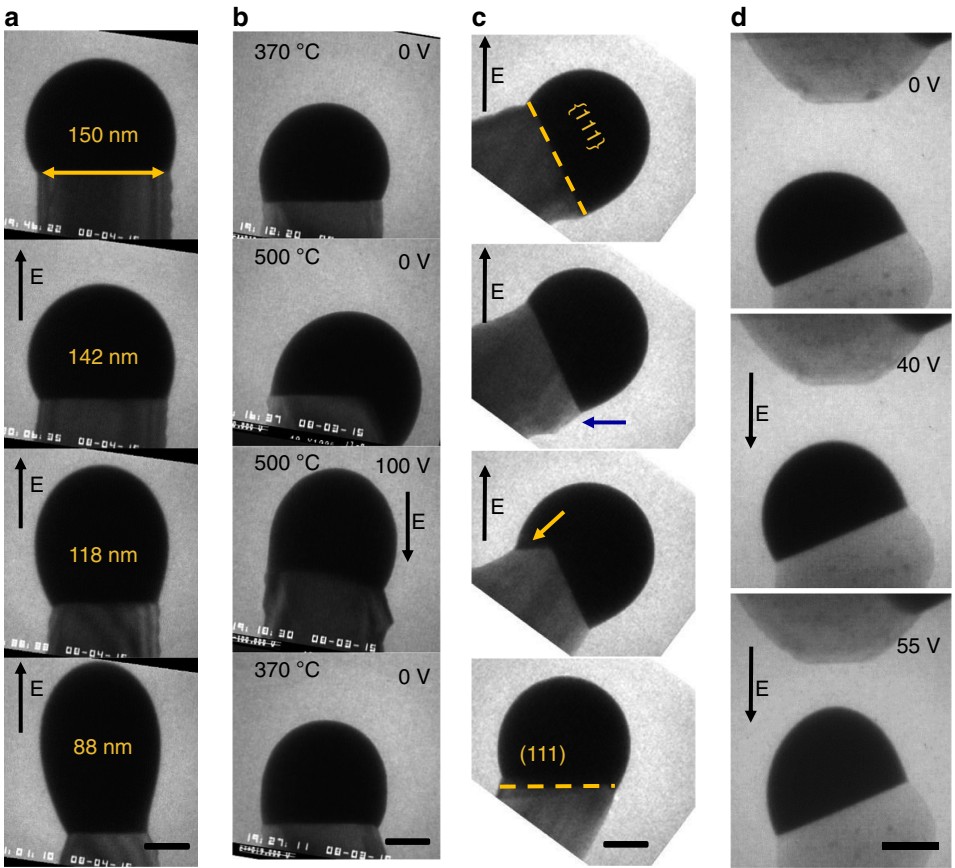

**Figure 2 | Nanowire growth controlled by electric field. (a)** Image sequence showing a nanowire growing at 480 °C and $1.2 \times 10^{-5}$ Torr $Si_2H_6$ in the [111] direction towards the CE, 800 nm distant, under an applied voltage of $-90$ V (E-field is $\sim 100$ V $\mu m^{-1}$). The progressively decreasing nanowire diameter is reported in each image. **(b)** Sequence in which a nanowire growing stably in the [111] direction at $\sim 1.6 \times 10^{-5}$ Torr and 500 °C is cooled to 370 °C (first image). Heating back to 500 °C destabilizes the droplet (second image). A field is applied and 'repositions' the droplet back to the original growth plane (third image) so growth continues without a kink. **(c)** Sequence showing a nanowire growing in a <111> direction at 510 °C and $\sim 1.6 \times 10^{-5}$ Torr of $Si_2H_6$. When $-52$ V is applied on the CE generating a field in the [111] direction (at $\sim 70°$ to the growth direction), the nanowire gradually kinks to this direction. **(d)** Sequence showing a droplet deformed towards an asperity on the CE (in this case, a previously grown nanowire that has lost its droplet). Scale bars, 50 nm; arrows represent the direction of the E-field inside the capacitor.

for the range accessible here (in Fig. 4e, $\gamma$ remains essentially constant for $E$ in the range 1.25–2.8 V nm$^{-1}$). This method allows us to measure $\gamma$ under growth conditions, and does not require any knowledge of the surface energy of the substrate on which the droplet is sitting. Knowledge of $\gamma$ is useful in improving growth models[15,19] and allowing more accurate simulation of growth under external stimuli.

## Discussion

To develop applications of electric field-directed nanowire growth, it is important to apply the technique within conventional growth reactors. The main differences between our experimental conditions and conventional reactor conditions are the gas pressure and the presence of a CE close to the growth substrate. Nanowire growth at standard reactor pressures in the mTorr to Torr range has been carried out under fields of similar strength without causing an electric breakdown[26], as expected by the Paschen law for silane[23]. Since pressure mainly changes the rate rather than the mechanism of nanowire growth, we, therefore, expect that at reactor pressures and in the applied field, phenomena such as diameter modulation and kinking will still take place as described above but over shorter time scales. However, the presence of an external CE in the immediate

proximity of the sample is not easily scalable to standard reactors. We, therefore, propose two modifications that can allow a field to be applied during growth. The first consists in growing nanowires on a patterned and polarizable substrate or on a nano-electromechanical structure[30–32] so that either the nanowires themselves, or nearby previously fabricated nanostructures, will act as the CE. This can in principle direct nanowire growth into complex networks, as shown in Supplementary Fig. 4a,b. A second approach is to exploit the strong electric field generated in the Debye sheath during conventional plasma enhanced chemical vapour deposition, Supplementary Fig. 4c. This has been successfully used for the alignment of carbon nanotubes[33–36]. With these approaches we believe that the use of electric fields to control nanowire growth could become feasible in conventional reactors (see also Supplementary Note 1).

The ability to control nanowire growth by an electric field suggests some intriguing possible applications. Diameter modulation can increase surface scattering of phonons, important for thermoelectric applications[37]. Controlled kinking could lead to the fabrication of three-dimensional structures with applications in electronics[7] and sensing[38]. Field-controlled kinking could perhaps even force nanowires growing laterally ('crawling') along the substrate[39] into vertical growth. Directing wires to kink towards each other, as in Supplementary Fig. 4a,b,

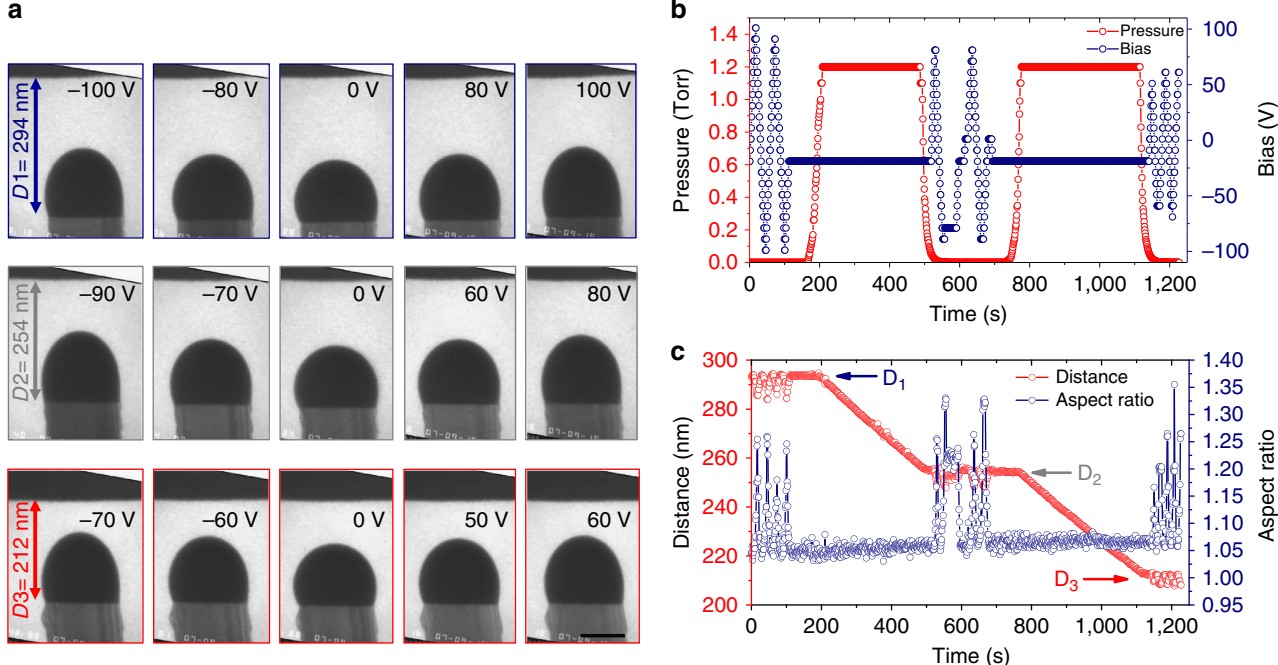

**Figure 3 | Droplet deformation induced by electric field.** A nanowire (W2, see Fig. 4 below) was grown at $\sim 500\,^{\circ}$C and $1.2 \times 10^{-5}$ Torr $Si_2H_6$, its growth interrupted at different distances from the CE and a field then applied. (**a**) Image sequence showing droplet shape at the voltages specified and at three different distances. The voltage range was chosen to obtain similar maximum deformation at each distance. Scale bar, 100 nm. (**b**) Pressure and voltage versus time. (**c**) Distance between nanowire and CE, and droplet aspect ratio ($b/a$) versus time. As expected, the voltage required to obtain a given deformation decreases as the droplet approaches the CE. Note that the distance $D_x$ is also slightly affected by the electric field due to electrostatic attraction between the cantilevers (see Methods).

may help in creating complex structures with electronic applications. An example is the X-junction geometries sought for devices that involve Majorana fermions[40,41]. Finally, in materials such as GaAs, it is known that the nanowire can grow with either a hexagonal or a cubic structure, with the choice determined by the contact angle[6]. This implies that it should be possible to form crystal phase heterostructures in III–V nanowires by controlling contact angle with field, as shown in Supplementary Fig. 5.

We have shown that an externally applied electric field can strongly affect the growth of nanowires via the VLS process. Real-time imaging provides a direct demonstration that the field controls growth by inducing changes in the droplet geometry, in particular the droplet position and the droplet–nanowire contact angle. An axial electric field can stabilize growth and change nanowire diameter reversibly. Breaking the symmetry by applying the field at an angle to the nanowire axis allows kinking in a desired direction. We have also shown that nanoscale electrodes, such as a previously grown nanostructure, can be used for local control of nanowire growth directions. Finally, the ability to apply an electric field allows measurement of the surface tension of the AuSi droplet under growth conditions, which, as well as being useful for modelling growth, allows prediction of the droplet shape in more general fields and geometries. We believe that the use of externally applied fields provide new opportunities for nanowire growth, with the ability to create structures that cannot be obtained through the global control of parameters such as temperature, gas composition and gas pressure. If it can be adapted to conventional growth reactors, electric field-enabled droplet engineering can provide exciting opportunities for enhancing the VLS technique to create a variety of complex, three dimensional nanowire-based structures.

## Methods

**Sample fabrication.** The cantilever loops were fabricated in a process similar to that described in Kallesøe et al.[22] and Alam et al.[42]. Silicon cantilever heaters were fabricated by etching the device layer of a silicon-on-insulator wafer using reactive ion etching[42]. The device layer was $4 \pm 0.5\,\mu$m in thickness, had a resistivity of $0.085\,\Omega$cm (Boron, $3 \times 10^{17}\,cm^{-3}$), and $<110>$ orientation with the cantilever sidewalls being the desired {111} planes. After fabrication, the cantilevers were etched in 28% KOH for 20 s at room temperature to planarize the {111} sidewalls. For growth, the cantilevers were Joule heated to 470–550 $^{\circ}$C by applying typically 28–35 mW (0.65–1 mA and 25–35 V) to each cantilever loop. This was supplied by two Keithley 2400 SourceMeters set up as constant current sources. The temperature was calibrated from nanowire growth rate using the method described in Alam et al.[42]. A third power supply controlled the voltage between the loops. The maximum voltage applied, 200 V, resulted in a leakage current below 1 $\mu$A.

Nanowire growth was performed in a Hitachi H-9000 ultrahigh vacuum TEM having a base pressure of $2 \times 10^{-10}$ Torr and a maximum pressure during imaging of $2 \times 10^{-5}$ Torr. This microscope is connected to a cluster of ultrahigh vacuum tools where metal deposition was carried out. The native oxide was removed from the cantilever surface by 10% hydrofluoric acid (HF) etchant solution for 2 min. The chip was then immediately transferred (within 2 min) to the TEM loadlock, where it was baked at 100–150 $^{\circ}$C under a tungsten lamp for 8 h to degas and remove moisture. It was then transferred under vacuum to a Knudsen cell Au evaporation system, thus maintaining an oxide-free surface. Less than 5 nm Au was deposited at a grazing angle (10–15$^{\circ}$) to the cantilever sidewalls to act as the VLS catalyst. The sample was then transferred under vacuum to the TEM where precursor gas ($Si_2H_6$) was introduced through a capillary tube. The temperature was then raised to 500 $^{\circ}$C in order to agglomerate Au into droplets and initiate VLS growth. Typical growth rates are in the range 5–15 nm min$^{-1}$ at temperatures of 470–520 $^{\circ}$C.

When a voltage is applied we measure a small change, $\Delta D$, in the distance between nanowire and cantilever. We suggest that this is due to electrostatic attraction between the two sides of the capacitor (rather than say an elongation of the nanowire pulled by its droplet) since $\Delta D$ is larger when the applied voltage is greater (Fig. 3b), that is, the electric field between the loops is greater, and not when the droplet deformation is larger (Fig. 3c), that is, when the local electric field between droplet and CE is greater. This accounts for the few per cent distance variation at each fixed $D_x$ in Fig. 3c.

We have evaluated the possibility of electron beam effects in these experiments. Changing the beam intensity does not result in a detectable change in the deformation or position of the droplet, or in the nanowire growth kinetics.

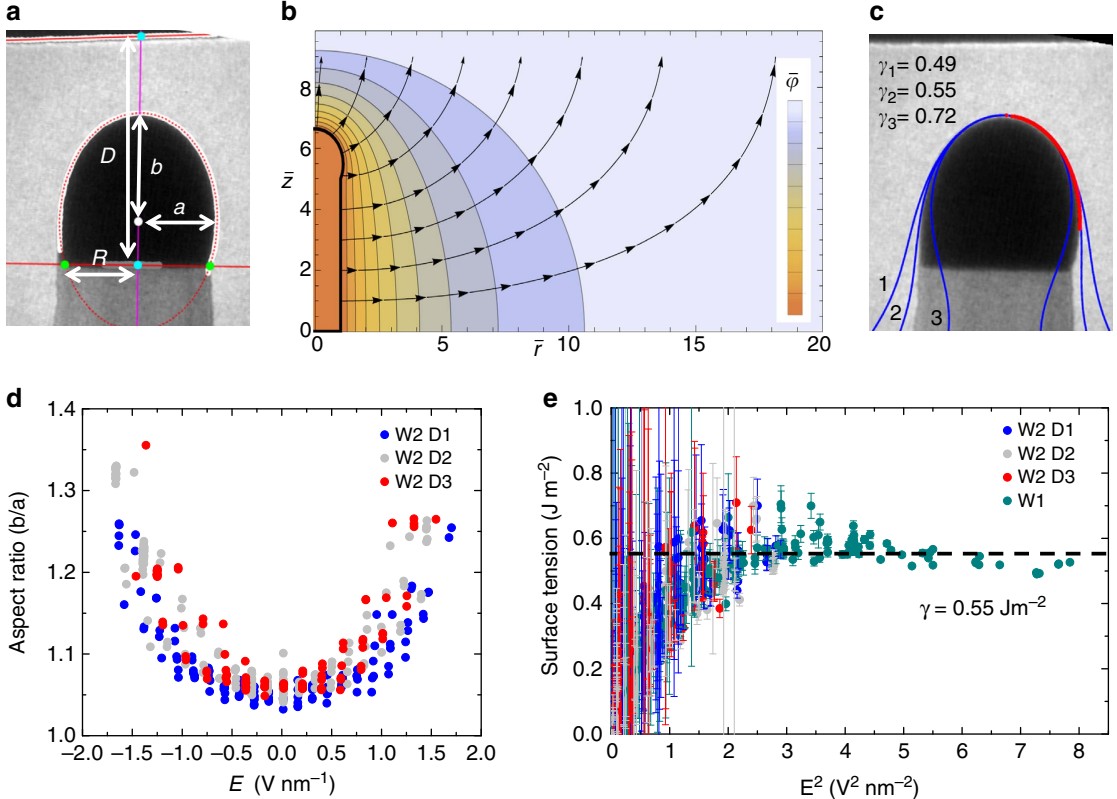

**Figure 4 | Calculation of surface tension.** (**a**) Example of automated fitting of $a$ and $b$ for a single frame of a movie. (**b**) Two dimensional diagram showing the calculated potential distribution and electric field lines around the nanowire. The nanowire and droplet surface are arbitrarily set to $\varphi = 0$ and the CE to $\varphi = V$; the system is solved in dimensionless form using the measured radius of the nanowire $R$ and potential $V$ to scale the system: $\bar{r} = r/R$, $\bar{z} = z/R$ and $\bar{\varphi} = \varphi/V$. (**c**) Calculated droplet shape for three different values of surface tension (0.49; 0.55; 0.72 J m$^{-2}$). The red highlight shows the portion of the surface used to evaluate the best fit, far enough from the nanowire interface to be relatively unaffected by the cross sectional shape. (**d**) Droplet aspect ratio versus calculated electric field at the apex of the droplet. The three data sets in Fig. 3 (W2: $D_1$, $D_2$, $D_3$) are superimposed. (**e**) Best fitting surface tension $\gamma$ versus square of electric field at the droplet apex. The three data sets in Fig. 3 and data from Fig. 1 (W1) are superimposed. The error bars account for the calculation artifact described in Methods. We choose error in $\gamma$ that scales as $1/(a/b - a_0/b_0)$, where $a_0/b_0$ is the measured aspect ratio of the droplet at zero-field. This captures the fact that deviations from the unperturbed geometry need to be large before their measurement becomes significant. The horizontal dashed line shows the value of surface tension obtained as error weight average of all data, $\gamma = 0.55$ J m$^{-2}$.

Furthermore, we do not observe measurable charging effects from the field: the droplet shape responds rapidly to changes in E-field direction (Supplementary Movie 1). Although the nanowires are not deliberately doped, we measured a resistivity of the order of $1\,\Omega$ cm[42], 3–4 orders of magnitude lower than intrinsic bulk Si and more than 12 orders of magnitude lower than SiO$_2$. This low resistivity can be explained by surface conduction or unintentional doping of Si due to experiments in the microscope involving other materials. For this reason our nanowires can be considered conductive and we can exclude the presence of charge accumulation due to the electron beam.

**Simulations.** The first step of the process of calculating the surface tension from the droplet deformation is to identify, using edge detection algorithms on thresholded images, the points in each image corresponding to the electrode and the droplet/nanowire and droplet/vacuum interfaces. These points were used to obtain the nanowire radius $R$, distance between nanowire and CE, and droplet geometry. From this, an idealized axisymmetric representation of the experimental geometry was created. The electric potential was calculated on this domain using the finite element method in *Mathematica*. The nanowire and droplet surface were arbitrarily set to $\varphi = 0$ and the CE to $\varphi = V$; the natural boundary condition $\hat{\mathrm{n}} \cdot \nabla \varphi = 0$ is applied everywhere else. We solve the system in dimensionless form using the measured $R$ and known $V$ to scale the system: $\bar{r} = r/R$, $\bar{z} = z/R$, $\bar{\varphi} = \varphi/V$.

$$\bar{\nabla}^2 \bar{\varphi} = \frac{\partial^2 \bar{\varphi}}{\partial \bar{z}^2} + \frac{1}{\bar{r}} \frac{\partial}{\partial \bar{r}} \bar{r} \frac{\partial \bar{\varphi}}{\partial \bar{r}} = 0 \tag{2}$$

Once the electric field distribution along the droplet has been found, it can be used to obtain the Maxwell stress distribution, which, for a conducting droplet, modifies

the Young–Laplace equation as follows:

$$\gamma \left( \frac{1}{R_1} + \frac{1}{R_2} \right) = \Delta P_0 + \frac{1}{2} \varepsilon_0 E_n^2 \tag{3}$$

We can take advantage of the additional symmetry at the apex to define the unknown constant $\Delta P_0$ in terms of the mean curvature $(1/R_1)|_{s=0} = (1/R_2)|_{s=0} = b/a^2$ and apex electric field $E_n^2|_{s=0}$ such that

$$\Delta P_0 = 2\gamma \frac{b}{a^2} - \frac{1}{2} \varepsilon_0 \, E_n^2 \big|_{s=0} \tag{4}$$

Here $s$ is the distance along the interface, starting at the apex of the droplet and following the droplet curvature. The principal curvatures can be recast in differential form as $1/R_1 = d\phi/ds$ and $1/R_2 = \sin\phi/r$ following Río *et al.*[43]. Equation 4, along with two geometric relationships, becomes the system:

$$\begin{aligned} \frac{d\phi}{ds} + \frac{\sin\phi}{r} &= 2\gamma \frac{b}{a^2} + \frac{1}{2} \varepsilon_0 E_n^2(z(s)) - \frac{1}{2} \varepsilon_0 \, E_n^2 \big|_{s=0} \\ \frac{dr}{ds} &= \cos\phi \\ \frac{dz}{ds} &= \sin\phi \end{aligned} \tag{5}$$

with initial conditions

$$\phi(0) = r(0) = z(0) = 0 \tag{6}$$

Non-dimensionalizing the system and boundary conditions according to the scales

used in Equation 2 gives

$$\frac{d\phi}{d\bar{s}} + \frac{\sin\phi}{\bar{r}} = 2\frac{\bar{b}}{\bar{a}^2} + \Omega\left(\frac{1}{2}\bar{E}_{\mathrm{n}}^2 - \frac{1}{2}\varepsilon_0\bar{E}_{\mathrm{n}}^2\big|_{s=0}\right)$$
$$\frac{d\bar{r}}{d\bar{s}} = \cos\phi \tag{7}$$
$$\frac{d\bar{z}}{d\bar{s}} = \sin\phi$$

with dimensionless Electric Bond Number $\Omega = V^2\varepsilon_0/\gamma R$ arising naturally. We find the best fit between the theoretical predictions (subscript $t$) and $N$ points from the best-fit ellipse (subscript $e$) by minimizing the penalty function with respect to $\Omega$

$$\Phi(\Omega) = \sum_{i=1}^{N}\sqrt{\left(\bar{r}_{t,i}(\Omega) - \bar{r}_{e,i}\right)^2 + \left(\bar{z}_{t,i}(\Omega) - \bar{z}_{e,i}\right)^2} \tag{8}$$

A more general penalty function is discussed in Río et al.[43], which accounts for rotation of the acquired profile; however, the orientation of our profiles have already been determined using the best-fit ellipse from the image processing step. Once $\Omega$ has been determined, the surface tension can be found directly from $\gamma = V^2\varepsilon_0/\Omega R$.

**Errors in the measurement of surface tension.** We find that the main source of error in the measurement of surface tension is the systematic error in determining the distance $D$ between the droplet tip and the CE. The CE is a microfabricated surface, viewed in projection and not perfectly flat. Since it is not always obvious in projection which part of the surface is nearest the droplet, we estimate that error in $D$ could be of the order of several tens of nm in a single data set, corresponding to a $\sim 20\%$ error in $\gamma$. Note that the true value of $D$ is always greater than or equal to the apparent value in the images. This error can be eliminated with the method shown in Fig. 3, obtaining data at several $D$ values between which the nanowire was grown by a known length. If $D_x$ were incorrect, the three $b/a$ versus $E$ curves would not coincide as they do in Fig. 4d.

A second source of error arises from the geometry-induced deformation of the droplet. The droplet is distorted because it sits on a hexagonal base, and exhibits a non-spherical cap shape ($b_0/a_0 = 1.05$) at zero field. Our fitting approach does not take into account this deviation from the ideal shape since this would require a more complex three-dimensional calculation. The program, therefore, yields fitted values of $\gamma$ that tend to zero as the field approaches zero, because a weak field can only distort a droplet if it has a very low surface tension. The correct value of $\gamma$ is only obtained if the non-spherical droplet distortion is overridden by electric field-induced deformation. Deviations from the unperturbed geometry need to be large before their measurement becomes significant. In order to account for this systematic error, we assign an error bar in $\gamma$ that scales as $1/(a/b - a_0/b_0)$.

**Data availability.** The data that support the findings of this study are available from the corresponding authors upon request.

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

## Acknowledgements

Supported by ERC Grant 279342: InSituNANO and EPSRC Grant EP/P005152/1 (F.P., S.H.), by the National Science Foundation CBET 1066573 and the Nano/Bio Interface Center through the National Science Foundation NSEC DMR08-32802 (M.M.N.), and the FTP Nano Live project funded by The Danish Council for Independent Research, Case No.10-083797 (S.B.A., K.M.). The authors acknowledge Jerry D. Tersoff and Haim H. Bau for fruitful discussions and Mark C. Reuter, Arthur W. Ellis and Eric Jensen for technical support.

## Author contributions

F.P. performed experiments and data analysis, S.B.A. performed experiments and fabricated growth substrates, M.M.N. performed the simulations, and K.M., S.H. and F.M.R. designed the experiments and coordinated the analysis. All authors contributed to writing the paper.

## Additional information

**Competing financial interests:** The authors declare no competing financial interests.

