## [Peer Review File · Nature Communications]

Reviewers' comments:

Reviewer #1 (Remarks to the Author):

The work of F. Panciera et al. reports on the growth control of silicon nanowires under the influence of an electric field. The experiments are carried out in-situ in a transmission electron microscope (TEM). To this end, vapor liquid solid (VLS) growth of silicon nanowires is observed while the electric field is being altered during growth. The authors show, that the shape of the Au-Si liquid catalyst particle is altered by the applied field without a change in composition and temperature of the catalyst droplet. They show that they can control the nanowire diameter and to a certain extent the nanowire growth direction reversibly by altering the electric field. Apart from this droplet engineering, the authors propose to use the method in order to extract the surface tension of the catalyst particle during nanowire growth. This is an important parameter that dictates VLS growth mechanism.

The paper is very well and clearly written. References are appropriate. The experiments are well designed and analyzed. Although field induced VLS nanowire growth has been reported previously, no field influenced in-situ growth has been reported up-to-date. The results are all novel and relevant to the fairly large nanowire growth and device community.

Due to its novelty and potential impact I recommend this manuscript for being published at Nature Communications after minor corrections:

- 1.- The authors should comment on the influence of the electron beam on the observed experiments. Can charging of the catalyst particle completely be disregarded? The nanowire is undoped, so can accumulated charge be drained to the cantilever structure?
- 2.- Please comment on the limitations of the field influenced growth under different range of pressure and field conditions. Although the authors point out, that experiments have been done with comparable fields in LPCVD conditions, many readers will be interested in the limitation in terms of impact ionization and breakdown. Where are the experiments situated in the Paschen curve for the process gases used? Please comment, when do you expect impact ionization to occur with the used gases.
- 3.- The error margins in graph 4e need to be shown/discussed?

Minor comment, Please recheck your references. Reference 17 is identical to 27.

Reviewer #2 (Remarks to the Author):

This article by Panciera et al. describes in situ TEM observations of droplet deformation in the presence of an electric field, on a customized MEMS platform for VLS inside of a TEM. The reviewer appreciates the complexity required to accomplish this integrated setup, and recommends this work for publication. The article is concise and easy to read, and presents interesting conclusions on the physics of droplet deformation, and how it relates to the electric field profile. The droplet geometry is modeled by electromagnetic simulations coupled to the Young-Laplace equation. This is of general interest to the nanowire growth community, and the data and conclusions are sound. The reviewer supports publication, and suggests the following minor points that could improve the discussion:

- 1) One of the major effects of growing the nanowires in the TEM is the lower pressures present in the vacuum system. Can the authors extrapolate their model parameters, to estimate the results in a normal VLS growth chamber? Also, this could be compared to VLS grown NWs via typical conditions, to see if the model predicts the changes in growth geometry based on the new understanding gained from this paper. By coupling this in situ TEM analysis to understanding

standard laboratory growth behavior, the results will be more impactful to the general community.

2) The majority of the discussion here is on deformation of the catalyst droplet. However, e-field effects may also affect diffusion across the liquid-solid interface, and solidification kinetics. Do the authors observe any differences in diffusion kinetics during solidification as a function of field strength?

3) Does the electric field from the TEM electron beam effect the growth dynamics? Was the current/dosage of the beam found to have an impact on the results?

4) The reviewer suggests adding a few more major review articles on semiconductor nanowire growth mechanisms summarizing the current state of the art in the field in the introduction (e.g. Adv. Mater. 26, 2137, 2014; Small 2, 700, 2006), as the physics of VLS nanowire growth have been widely studied by many groups, including droplet geometry effects, surface diffusion of catalyst particles, growth of hierarchical kinked structures etc. This will aid in the discussion for the general reader of Nature Communications, who may not be familiar with the depth of existing knowledge in the field.

Reviewer #3 (Remarks to the Author):

The authors demonstrate a unique method for controlling the synthesis of semiconductor nanowires using electric fields. The growth is conducted using in-situ TEM in order to fully understand the processes occurring. The authors also use this method to extract values for the droplet surface tension under synthesis conditions. Overall, the work is very interesting and the concept is very novel, and potentially useful for researchers working with nanowire growth. However, the amount of data shown seems rather limited to support the claims made by the authors, especially on the generality and usefulness of this method. I suggest the authors answer the following:

- The authors say that diameter modulation is possible, since the droplet deformation caused by the electric field is reversible. However, I could not find that they include any data supporting this claim. An experiment showing diameter modulation should be included

- Although the authors say that contact angle affects how the nanowires grow, in their experiments they seem only to address diameter and position of the droplet. Does the change in contact angle shown in for example Figure 1 change the way the nanowire grows?

- The experiments are interesting for demonstrating qualitative behavior, it is unclear whether this will have practical use. For example, it seems the nanowire can only grow for a short time since the distance to the counterelectrode is short. Can the authors comment? How practical is this method for actual nanowire engineering?

- Much space is used for speculating about the applications of this technique, for example on page 11. However I have difficulty understanding how any of the situations on page 11 could be realized. Perhaps the authors could explain, or better, demonstrate the use of this technique for at least one of the concepts listed here in order to show that the technique is robust

- On a related note, the authors say that transfer of this technique to conventional reactors should be feasible. I think this requires further discussion and it is not clear how this would be possible

In summary, this work is novel and worthy of publication if the authors can provide a bit more information to support their claims

Editorial comments

The manuscript has been modified in order to meet the requirements for Nature Communications.

We added an abstract and headings for the different sections of the text.

Reviewer #1 (Remarks to the Author):

The work of F. Panciera et al. reports on the growth control of silicon nanowires under the influence of an electric field. The experiments are carried out in-situ in a transmission electron microscope (TEM). To this end, vapor liquid solid (VLS) growth of silicon nanowires is observed while the electric field is being altered during growth. The authors show, that the shape of the Au-Si liquid catalyst particle is altered by the applied field without a change in composition and temperature of the catalyst droplet. They show that they can control the nanowire diameter and to a certain extent the nanowire growth direction reversibly by altering the electric field. Apart from this droplet engineering, the authors propose to use the method in order to extract the surface tension of the catalyst particle during nanowire growth. This is an important parameter that dictates VLS growth mechanism.

1.1 The paper is very well and clearly written. References are appropriate. The experiments are well designed and analyzed. Although field induced VLS nanowire growth has been reported previously, no field influenced in-situ growth has been reported up-to-date. The results are all novel and relevant to the fairly large nanowire growth and device community. Due to it's novelty and potential impact I recommend this manuscript for being published at Nature Communications after minor corrections:

We appreciate these positive comments.

1.2 The authors should comment on the influence of the electron beam on the observed experiments. Can charging of the catalyst particle completely be disregarded? The nanowire is undoped, so can accumulated charge be drained to the cantilever structure?

Answer to 1.2: Beam effects are always one of our concerns during in-situ experiments and we take care to periodically observe nanowires (NWs) that are not constantly exposed to the beam in order to determine any beam-induced difference. For the specific case of Si NWs, we have not observed any major effect of the beam on growth. Since electric field effects on growth have not been studied before in situ, we therefore paid particular attention to any possible interaction between the E field and the beam. Changing the irradiation intensity did not make an observable difference in the deformation of the droplet or the growth of wires. An experimental evidence of the absence of significant charging effects is represented by the rapid response of the droplet shape to the change of E field direction in movie 1. The NWs are not deliberately doped. However, we measured a resistivity of the order of $1 \text{ } \Omega\text{cm}$ [¹], 3-4 orders of magnitude lower

than intrinsic bulk Si and more than 12 orders of magnitude lower than SiO₂. This low resistivity can be explained by surface conduction or unintentional doping of Si due to other experiments (such as GaAs growth) carried out in the same microscope. For this reason our nanowires can be considered conductive and we can exclude the presence of charge accumulation due to electron beam.

We added a sentence explaining this point in the Methods section.

1.3 Please comment on the limitations of the field influenced growth under different range of pressure and field conditions. Although the authors point out, that experiments have been done with comparable fields in LPCVD conditions, many readers will be interested in the limitation in terms of impact ionization and breakdown. Where are the experiments situated in the Paschen curve for the process gases used? Please comment, when do you expect impact ionization to occur with the used gases.

Answer to 1.3: This is an important point and we thank the reviewer for addressing it. Growth of nanowires has been done in a DC electric fields comparable with the field used in our work and at standard CVD pressure (25 V/ μm and 350 mTorr [²]) without igniting a plasma. In fact, the Paschen curve for SiH₄ (Fig. 1) shows that, regardless of the pressure and the interelectrode distance, the breakdown voltage is always higher than 500 V. So in our case and in most cases relevant for applications, we are clearly far from conditions where a plasma can be generated. However, in a PECVD reactor, a plasma can be sustained by a RF source or by a DC field assisted by external source of electrons. We discuss possible applications of our technique to the case of PECVD in the answer to point 3.4 below.

We added a sentence at page 2 stating that plasma ignition requires at least 500 V for silane. We also added a paragraph in the Discussion section in which we discuss the transfer of our technique to a conventional reactor and the opportunity provided by the use of PECVD.

Figure 1. Measured Paschen law for silane, hydrogen, nitrogen and argon [³]. p is the pressure and L is the interelectrode distance.

1.4 The error margins in graph 4e need to be shown/discussed?

Answer to 1.4: As the error on the surface tension calculation is dominated by the geometry-induced systematic error, we did not originally plot it on the graph but only discussed it in the text. However we agree that adding error bars would help the reader.

We modified Figure 4e as shown below and in the figure caption we now describe how the error bars are calculated.

Figure 2. Best fitting surface tension vs. square of electric field at the droplet apex. The dataset in Figure 1 (W1) and the three datasets of Figure 3 (W2 D1,2,3) are superimposed. The error bars account for the calculation artifact described in Supporting Information Section 4. We choose error in γ that scales as $1/(a/b - a_0/b_0)$, where a_0/b_0 is the measured aspect ratio of the droplet at zero-field. This captures the fact that deviations from the unperturbed geometry need to be large before their measurement becomes significant. The horizontal dashed line shows the value of surface tension obtained as error weight average of all data, $\gamma = 0.55 \text{ J/m}^2$.

1.5 Minor comment, Please recheck your references. Reference 17 is identical to 27.

Answer to 1.5: Thanks, we corrected the reference numbers.

Reviewer #2 (Remarks to the Author):

This article by Panciera et al. describes in situ TEM observations of droplet deformation in the presence of an electric field, on a customized MEMS platform for VLS inside of a TEM. The reviewer appreciates the complexity required to accomplish this integrated setup, and recommends this work for publication. The article is concise and easy to read, and presents interesting conclusions on the physics of droplet deformation, and how it relates to the electric field profile. The droplet geometry is modeled by electromagnetic simulations coupled to the Young-Laplace equation. This is of general interest to the

nanowire growth community, and the data and conclusions are sound. The reviewer supports publication, and suggests the following minor points that could improve the discussion:

2.1 One of the major effects of growing the nanowires in the TEM is the lower pressures present in the vacuum system. Can the authors extrapolate their model parameters, to estimate the results in a normal VLS growth chamber? Also, this could be compared to VLS grown NWs via typical conditions, to see if the model predicts the changes in growth geometry based on the new understanding gained from this paper. By coupling this in situ TEM analysis to understanding standard laboratory growth behavior, the results will be more impactful to the general community.

Answer to 2.1: As discussed in the answer to 1.3 above, NW growth under electric field can also be carried out at standard reactor pressure. To our knowledge, only one example of NW growth directed by E-field is reported in literature [2]. In that case, several μm long NWs converge to a point-like CE by bending. The mechanism is different from what we propose here so direct comparison is not straightforward. However, over the range of pressure accessible in situ, it is known that growth mechanisms are consistent: pressure only changes the rate of the processes. We therefore expect that phenomena described here such as kinking will still take place at reactor pressures but over shorter time scales.

We added a sentence about controlling NW kinking and diameter modulation in conventional reactor conditions in the Discussion (page 12).

2.2 The majority of the discussion here is on deformation of the catalyst droplet. However, e-field effects may also affect diffusion across the liquid-solid interface, and solidification kinetics. Do the authors observe any differences in diffusion kinetics during solidification as a function of field strength?

Answer to 2.2: This is an interesting observation and we are aware that surface diffusion may be directed by strong electric fields [4,5]. However, we did not observe any measurable difference in growth rate when the field was applied. Current understanding of Au catalyzed Si NW growth suggests that Si diffusion at wire sidewalls or across the liquid/solid interface does not control growth rates. Moreover, because of the conductivity of NW and droplet (see 1.2 above) the electric field does not penetrate so can not affect the diffusion within the liquid or at the liquid/solid interface. Electric field induced surface diffusion may play an important role in the growth III-V nanowires where the group III element is assumed to be highly mobile on the NW sidewalls. It would be interesting to explore this possibility.

2.3 Does the electric field from the TEM electron beam effect the growth dynamics? Was the current/dosage of the beam found to have an impact on the results?

Answer to 2.3: Please see point 1.3 above.

2.4 The reviewer suggests adding a few more major review articles on semiconductor nanowire growth mechanisms summarizing the current state of the art in the field in the introduction (e.g. Adv. Mater. 26, 2137, 2014; Small 2, 700, 2006), as the physics of VLS nanowire growth have been widely studied by many groups, including droplet geometry effects, surface diffusion of catalyst particles, growth of hierarchical kinked structures etc. This will aid in the discussion for the general reader of Nature Communications, who may not be familiar with the depth of existing knowledge in the field.

Answer to 2.4: Thanks for suggesting that we provide additional background via review papers. We have added them as refs #3 and #4.

Reviewer #3 (Remarks to the Author):

The authors demonstrate a unique method for controlling the synthesis of semiconductor nanowires using electric fields. The growth is conducted using in-situ TEM in order to fully understand the processes occurring. The authors also use this method to extract values for the droplet surface tension under synthesis conditions. Overall, the work is very interesting and the concept is very novel, and potentially useful for researchers working with nanowire growth. However, the amount of data shown seems rather limited to support the claims made by the authors, especially on the generality and usefulness of this method. I suggest the authors answer the following:

3.1 The authors say that diameter modulation is possible, since the droplet deformation caused by the electric field is reversible. However, I could not find that they include any data supporting this claim. An experiment showing diameter modulation should be included

Answer to 3.1: We agree that more experimental evidence showing diameter modulation should be included and we added the following figure and comments to the SI.

(added to SI text) Figure 3 shows the effect of voltage modulation on the diameter of a nanowire during growth. When the droplet is stretched, the diameter progressively decreases in order to reestablish the equilibrium contact angle. After removing the electric field, the droplet and the diameter of the wire return to the initial shape and size. This shows that the diameter change is reversible and that field can be used to modulate nanowire cross section.

Figure 3. Nanowire growing at 480°C and 1.5×10^{-5} Torr Si_2H_6 under an applied voltage between 0 and 110 V. **a** Image sequence showing droplet deformation and diameter change. **b** Wire diameter (black) and applied voltage (blue) vs. time. The small diameter fluctuations uncorrelated with the V variation are caused by the sidewall saw-tooth geometry [6]. Colored circles identify the data corresponding to images in **a**.

3.2 Although the authors say that contact angle affects how the nanowires grow, in their experiments they seem only to address diameter and position of the droplet. Does the change in contact angle shown in for example Figure 1 change the way the nanowire grows?

Answer to 3.2: The change of contact angle could affect the growth in different ways. We show the effect on the diameter and position of the droplet, but we also expect an additional effect for certain III-V semiconductors (Figure 4). For GaAs, it has been shown that the atomic arrangement itself (wurtzite versus zinc blende structure) is determined by the nanowire/droplet contact angle [7]. The longitudinal deformation of the catalyst droplet causes a change in contact angle so in principle could induce switching between phases. We added Figure 4 to the SI to include this effect.

Figure 4. Schematics illustrating the concept of phase switching in GaAs by changing the droplet/wire contact angle with an electric field.

3.3 The experiments are interesting for demonstrating qualitative behavior, it is unclear whether this will have practical use. For example, it seems the nanowire can only grow for a short time since the distance to the counterelectrode is short. Can the authors comment? How practical is this method for actual nanowire engineering?

Answer to 3.3: We agree that in our current setup, NWs can only grow for a short time due to the small distance between the wire tip and the counterelectrode. This configuration was designed in order to carry out the experiment in the TEM where strict limitations are imposed by the technique (transparency of the sample, space and voltage limitations). We agree that the TEM realization does not itself permit practical application, so we have speculated on the use of different geometries that could allow this. In the following answer we present some examples.

3.4 Much space is used for speculating about the applications of this technique, for example on page 11. However I have difficulty understanding how any of the situations on page 11 could be realized. Perhaps the authors could explain, or better, demonstrate the use of this technique for at least one of the concepts listed here in order to show that the technique is robust

Answer to 3.4: We illustrate three methods below that could allow implementation in a conventional reactor. We are happy to have the opportunity to expand on these ideas, and present below a longer description and schematic illustration. We added this figure to the SI, changed the SI text and referred to this material in the Discussion section of the main text.

- a) Directing NW growth by an electric field can be used to self-assemble nanowires into complex nanostructures. In Figure 5.a we show an example of how NWs can be grown

on a patterned substrate that allows the polarization of a row or group of nanowires. The electrostatic attraction between wires is used to form X or Y shaped nanostructures depending on the initial wire of locations. Self-assembly of nanowire networks has been sought as platform for future advance electronics [8] including Majorana-fermion-based devices [9].

- b) In a similar way NWs can be grown on NEMS structures integrated on Si wafers [10,11,12]. The nanostructure can be then polarized in order to direct the growth (Fig. 5.b)
- c) It is well documented that growth of carbon nanotubes (CNT) in conventional PECVD reactors leads to CNT alignment due to the electric field generated in the Debye sheath (DS) [13,14,15,16]. The same concept can be applied to generate an electric field during NW growth. The advantage of using a plasma is that the E- field around the NWs is not determined by the distance and potential difference between NWs and CE, but by the thickness of the DS and the difference between the plasma potential (V_p) and the wire potential (V_d) (see Figure 5.c). Typical DS thicknesses are in the order of a few μm to a few mm. The DS thickness as well as V_p can be tuned by changing V_d and the reactor geometry. This method would allow us to generate an electric field of several $\text{V}/\mu\text{m}$ oriented parallel to the wire growth. As we pointed out in the answer to 1.3, a DC plasma requires at least 500 V, but it can be sustained at lower voltage by means of a RF source or by a source of electrons.

Figure 5. a Sequence of schematics showing the formation of complex NW structures by applying a voltage between NWs grown on patterned electrodes. The E-field induces the kinking of the wires and the assembly of X (for slightly misaligned wires) or Y (for aligned wires). **b** Sequence of schematics showing the growth of NWs on polarized NEMS structures. **c** Schematic showing the growth of nanowires in a PECVD reactor. The electric field generated at the Debye sheath (DS) can be used to deform the catalyst droplet.

3.5 On a related note, the authors say that transfer of this technique to conventional reactors should be feasible. I think this requires further discussion and it is not clear how this would be possible

Answer to 3.5: The main differences between our experimental conditions and a conventional reactor conditions are the source gas pressure and the distance to the counterelectrode. In the answer to 1.3 we show that NW growth at standard reactor pressure can be carried out under high electric field without causing an electric breakdown. In answer 3.4 we show two methods that do not require the presence of an external counterelectrode in the immediate proximity of the sample and that are compatible with standard reactors. With these approaches, we believe that the transfer of our technique to conventional reactors will be feasible and we anticipate future progress in this direction.

In summary, this work is novel and worthy of publication if the authors can provide a bit more information to support their claims

We appreciate the referee's comments and hope that the responses have clearer path to the practical use of fields to modify nanowire growth.

References

-
- ¹ Alam, S. B., Panciera, F., Hansen, O., Mølhave, K., Ross F. M. Creating new VLS silicon nanowire contact geometries by controlling local contact surface temperature *Nano Letter* **15**(10), 6535-6541 (2015).
 - ² Englander, O., Christensen, D., Kim, J., Lin, L. and Morris, S.J., Electric-field assisted growth and self-assembly of intrinsic silicon nanowires. *Nano letters*, **5**(4), 705-708 (2005).
 - ³ Maessen, K. M. H., Ph. D. Thesis, Utrecht University, Utrecht, the Nederland (1988).
 - ⁴ Gault, B., Danoix, F., Hoummada, K., Mangelinck, D. and Leitner, H., Impact of directional walk on atom probe microanalysis. *Ultramicroscopy*, **113**, 182-191 (2012).
 - ⁵ Gill, V., Guduru, P.R. and Sheldon, B.W., Electric field induced surface diffusion and micro/nano-scale island growth. *International Journal of Solids and Structures*, **45**(3), 943-958 (2008).

-
- ⁶ Ross, F.M., Tersoff, J., Reuter, M.C., Sawtooth faceting in silicon nanowires. *Physical review letters*, **95**(14), 146104 (2005).
- ⁷ Jacobsson, D. *et al.* Interface dynamics and crystal phase switching in GaAs nanowires *Nature* **531**, 317-322 (2016).
- ⁸ Dai, X., Dayeh, S.A., Veeramuthu, V., Larrue, A., Wang, J., Su, H. and Soci, C., Tailoring the vapor–liquid–solid growth toward the self-assembly of GaAs nanowire junctions. *Nano letters*, **11**(11), 4947-4952 (2011).
- ⁹ Car, D., Wang, J., Verheijen, M.A., Bakkers, E.P. and Plissard, S.R., Rationally Designed Single-Crystalline Nanowire Networks. *Advanced Materials*, **26**(28), 4875-4879 (2014).
- ¹⁰ Islam, M.S., Sharma, S., Kamins, T.I. and Williams, R.S., Ultrahigh-density silicon nanobridges formed between two vertical silicon surfaces. *Nanotechnology*, **15**(5), L5 (2004).
- ¹¹ Sharma, S., Kamins, T.I., Islam, M.S., Williams, R.S. and Marshall, A.F., Structural characteristics and connection mechanism of gold-catalyzed bridging silicon nanowires. *Journal of crystal growth*, **280**(3), 562-568 (2005).
- ¹² Chaudhry, A., Ramamurthi, V., Fong, E. and Islam, M.S., Ultra-low contact resistance of epitaxially interfaced bridged silicon nanowires. *Nano letters*, **7**(6), 1536-1541 (2007).
- ¹³ Ren, Z.F., Huang, Z.P., Xu, J.W., Wang, J.H., Bush, P., Siegal, M.P. and Provencio, P.N., Synthesis of large arrays of well-aligned carbon nanotubes on glass. *Science*, **282**(5391), 1105-1107 (1998).
- ¹⁴ Chhowalla, M., Teo, K.B.K., Ducati, C., Rupesinghe, N.L., Amaratunga, G.A.J., Ferrari, A.C., Roy, D., Robertson, J. and Milne, W.I., Growth process conditions of vertically aligned carbon nanotubes using plasma enhanced chemical vapor deposition. *Journal of Applied Physics*, **90**(10), 5308-5317 (2001).
- ¹⁵ Hofmann, S., Csanyi, G., Ferrari, A.C., Payne, M.C. and Robertson, J., Surface diffusion: the low activation energy path for nanotube growth. *Physical review letters*, **95**(3), 036101 (2005).
- ¹⁶ Hofmann, S., Ducati, C., Robertson, J. and Kleinsorge, B., Low-temperature growth of carbon nanotubes by plasma-enhanced chemical vapor deposition. *Applied Physics Letters*, **83**(1), 135-137 (2003).

REVIEWERS' COMMENTS:

Reviewer #1 (Remarks to the Author):

The authors have satisfactorily addressed all issues raised by the reviewers. In-depth answers are given to each point in the response letter. In my opinion, the manuscript and supplementary information was enriched with these modifications. This specially holds for the section dealing with the applicability of their approach using conventional nanowire growth conditions. I recommend to publish this excellent, novel and highly relevant work in Nature Communications.

Reviewer #2 (Remarks to the Author):

The authors have satisfactorily addressed the previous concerns. I support this manuscript for publication.

Reviewer #3 (Remarks to the Author):

All comments from the previous round of revision have been addressed. in my opinion this manuscript is now suitable for publication.